# Quantitative Retention (Structure)–Activity Relationships in Predicting the Pharmaceutical and Toxic Properties of Potential Pesticides

**DOI:** 10.3390/molecules27113599

**Published:** 2022-06-03

**Authors:** Małgorzata Janicka, Anna Śliwińska

**Affiliations:** 1Department of Physical Chemistry, Faculty of Chemistry, Institute of Chemical Science, Maria Curie-Skłodowska University, 20-031 Lublin, Poland; 2Doctoral School of Quantitative and Natural Sciences, Maria Curie-Skłodowska University, 20-031 Lublin, Poland; aea.sliwinska@gmail.com

**Keywords:** lipophilicity, micellar chromatography, pesticides, QRARs, QSARs

## Abstract

The micellar liquid chromatography technique and quantitative retention (structure)–activity relationships method were used to predict properties of carbamic and phenoxyacetic acids derivatives, newly synthesized in our laboratory and considered as potential pesticides. Important properties of the test substances characterizing their potential significance as pesticides as well as threats to humans were considered: the volume of distribution, the unbonded fractions, the blood–brain distribution, the rate of skin and cell permeation, the dermal absorption, the binding to human serum albumin, partitioning between water and plants’ cuticles, and the lethal dose. Pharmacokinetic and toxicity parameters were predicted as functions of the solutes’ lipophilicities and the number of hydrogen bond donors, the number of hydrogen bond acceptors, and the number of rotatable bonds. The equations that were derived were evaluated statistically and cross-validated. Important features of the molecular structure influencing the properties of the tested substances were indicated. The QSAR models that were developed had high predictive ability and high reliability in modeling the properties of the molecules that were tested. The investigations highlighted the applicability of combined chromatographic technique and QS(R)ARs in modeling the important properties of potential pesticides and reducing unethical animal testing.

## 1. Introduction

Pesticides are very important substances in the modern world. They help to increase the efficiency of agricultural production and food processing by protecting crops against bacteria, fungi and molds, insects, rodents, and weeds. Since pesticides are used in the countryside, in forests, and in cities, people are constantly exposed to contact with these substances in their diets [1,2,3]. Although scientists do not have a full understanding of the health effects of pesticide residues, there is no doubt that the use of these substances must be limited and controlled. As new pesticide-active compounds are developed, it is vitally important to be able to predict their properties, their pharmacokinetics, and toxicities at the earliest stage of the research. Although modern science makes it possible to predict in silico the properties of substances only on the basis of their molecular structure, the results of these calculations rarely are highly reliable, and they generally require experimental verification. To avoid highly unethical and costly animal testing, alternative techniques in combination with in silico modeling can be used to predict the properties of drug-like or pesticide-like compounds in screening [4,5].

Reversed-phase liquid chromatography (RPLC), both planar and column, is a technique commonly used to assess the lipophilic properties of bioactive organic substances [6,7,8]. Chromatography with stationary phases that imitate biological partitioning systems, such as an artificial membrane, phases with immobilized lipids, albumin, cholesterol, ceramides, or liposomes, allows the prediction of the lipophilic properties [6,7,8,9,10] as well as the behaviors of solutes in real biological systems, such as bound to serum albumin, skin permeation, blood–brain barrier permeability, intestinal absorption, the concentration of unbound form in the blood, and others [11,12,13,14,15,16]. Similar possibilities are offered by micellar liquid chromatography (MLC) using surfactants as components of the mobile phase. MLC is a mode of conventional RPLC using a surfactant solution above the critical micellization concentration (*cmc*) in the mobile phase. Under these conditions, the micelles form the so-called micellar pseudophase in the bulk phase. The surrounding bulk water or aqueous-organic mixture contains surfactant monomers in a concentration approximately equal to the *cmc*. Surfactant monomers modify the surface phase as a result of the hydrophobic interactions between the tail of the surfactant and the alkyl chain. Molecular interactions present in this system, i.e., solute association with the polar head of the surfactant, solute penetration into the micelle core, and solute interactions with adsorbed surfactant and alkyl chains, affect retention by three different equilibria, which are (1) the solute distribution between the micelle (micellar pseudophase) and the bulk phase, (2) the solute partition between the stationary phase modified by the surfactant and the bulk phase, and (3) the direct transfer of solute molecules between the surfactant-modified surface and the micelle [17,18,19,20,21].

Several theories have been developed that describe the retention in MLC, i.e., the effect of the concentration of the surfactant in the effluent on the retention of the solute. Foley’s equation [22] is best known in lipophilicity studies, and according to Foley, the following relationship exists between the retention parameter, *k*, and the concentration of the surfactant in the effluent:(1)$$\frac{1}{k}=\frac{1}{k_{m}}+\frac{K_{\mathrm{AM}}}{k_{m}}\left[M\right]$$
where [*M*] is the total concentration of surfactant in the mobile phase minus *cmc*, *K*_AM_ is the constant that describes solute–micelle binding, and *k*_m_ is the solute retention parameter at zero micellar concentration, i.e., at surfactant monomer concentration equal to *cmc*. The *K*_AM_ and *k*_m_ parameters can be evaluated from the slope and intercept of experimental 1/*k* vs. [*M*] relationships. Equation (1) describes a linear dependence with decreasing retention as the micelle concentration increases. This equation is valid for aqueous solutions of a surfactant or mobile phases with the same concentrations of the organic modifier. The micellar retention parameter, log *k*_m_, is considered analogous to the log *k*_w_ value evaluated in RPLC. Thus, this parameter is considered a lipophilicity descriptor, and Equation (1) is a simple way to achieve the indirect determination of the lipophilic properties of compounds. It is postulated that retention in micellar chromatography depends on the hydrophobic (lipophilic), electronic, and steric features of the compounds in a similar way as many pharmacokinetic phenomena. An additional similarity is indicated by the fact that the phospholipids, cholesterol, fatty acids, and triglycerides that are present in the extracellular and intracellular fluids also form micelles with proteins.

In the studies, 15 carbamic and phenoxyacetic acids derivatives (Table 1), newly synthesized in our laboratory, and considered potential pesticides, were investigated using the column micellar liquid chromatography technique. As solutes lipophilicity descriptors there were applied *k*_m_ and *K*_AM_ values, calculated form Equation (1) [23]. Pharmacokinetic and toxicity parameters were predicted as functions of the solutes’ lipophilicities (QRARs model) or lipophilicity and the number of hydrogen bond donors (*HBD*), the number of hydrogen bond acceptors (*HBA*), and the number of rotatable bonds (*NRB*) (QSARs model).

## 2. Results

The retention parameters *k*_m_ and *K*_AM_ values are presented in Table 1. The relationship between the values was checked, and the following rectilinear relationship was obtained:log *k*_m_ = 0.585(0.102) + 0.932(0.044) log *K*_AM_(2)

*n* = 15; *s* = 0.1612; *R* = 0.9861; *R*_adj._ = 0.9849; *F* = 456; *p* = 0.000000.

The above confirms that both micellar parameters, i.e., log *k*_m_ and log *K*_AM_, could be used as alternative descriptors of the lipophilicities of compounds.

The physicochemical, pharmacokinetic, and toxicity parameters of the compounds (Table 2) are as follows: the logarithm of the partition coefficient (log *P*) in the *n*-octanol/water system, the number of hydrogen bond donors (*HBD*), acceptors (*HBA*), and rotatable bonds (*NRB*), molar weight (*MW*), topological polar surface area (*TPSA*) [24], the volume of distribution in the body (*V*_d_) [25], the fraction unbonded in a brain (*f*_u, brain_), in plasma (*f*_u, plasma_) [26], and pharmacokinetic parameters describing blood–brain distribution (log *BB*) [26,27,28,29], the rate of permeation from aqueous solutions through skin (log *K*_p_) [30,31], skin–water partition coefficient (log *K*_sc_) describing dermal absorption from aqueous solutions [32,33], the rate of permeation through cell (log *K*_w/cell_) [34], partitioning between water and serum albumin (log *P*_w/HSA_), and binding to human serum albumin (log *K*_HSA_) [10,35,36,37], partitioning between water and plant’ cuticles (log *P*_w/pc_) [38], and the dose causing the death of 50% of the group of mice tested after oral administration (*LD*_50_) [39,40]. These parameters describe important properties of the test substances and provide information about their potential applications as pesticides as well as potential threats to humans [41,42].

## 3. Discussion

Based on the anticipated in silico parameters (Table 2), it is important to note that the substances that were tested met the basic requirements formulated by Lipinski as the “rule of five” [43,44]: lipophilicities expressed as log *P* values are not greater than 5 (with the exception of substance no. 13); molecular weights are not greater than 500 g/mole; numbers of hydrogen bonds acceptors are not greater than 10, and the numbers of hydrogen bond donors are not greater than 5. In addition, the topological polar surface areas are below 90 Å^2^, and the number of rotatable bonds is in the range of 2–5. The compounds have moderate in silico predicted *V*_d_ values (*V*_d_ < 7 L/kg), indicating that they do not accumulate to a significant extent in fat tissue. The highest values of *V*_d_ were observed for compounds no. 12 and no. 13, the most lipophilic among all of the compounds that were tested. The values of log *K*_HSA_, which describe binding to human serum albumins, were in the range of 3.98–5.53, whereas the log *P*_w/HSA_ parameters that characterized the solute partitioning between water and serum albumin were in the range of 0.200–1.823. Xenobiotics bound to plasma proteins are not active because they are not able to cross membranes and permeate the site of action nor bind to receptors. The binding to serum albumins affects the concentration of the unbonded forms of the substance in serum, and small values of free fractions are preferable in order to prevent possible side effects.

Parameters that have negative values of log *BB* or values close to zero (in the range of −0.275 to −0.513) suggest that the compounds that were tested will not be able to penetrate into the brain, and neurotoxicity will be diminished. Of course, low levels of penetration into the brain are desirable. Compounds no. 12 and 13 had positive values of log *BB*, and they had the lowest unbonded fractions in the brain (*f*_u, brain_). Compounds no. 12 and 13, which had the highest log *K*_HSA_ and log *P*_w/HSA_ values, also had the lowest unbonded fractions in plasma (*f*_u, plasma_). According to the parameters calculated *in silico*, these substances are characterized by the highest rate of permeation through the skin (log *K*_p_), dermal absorption (log *K*_sc_), and partition between water and plant cuticle (log *K*_w/cell_), as well as the lowest rate of permeation through cells (log *P*_w/pc_). Substance no. 14 is the most toxic, poorly bound with albumins, and its concentration in the unbound form in the brain and serum was the highest among the substances that were tested, even though the value of log *BB* was negative.

When considering different parameters (e.g., chromatographic retention) as lipophilicity descriptors, they should be compared with the log *P* values that describe solute partitioning between *n*-octanol and water. In our studies, the relationships between the chromatographic and partition coefficient log *P* are presented in Figure 1. In both cases, i.e., for log *k*_m_ and log *K*_AM,_ the separate relationships for group I (carbamic acid derivatives) and group II (phenoxyacetic acid derivatives) were obtained with very good linearity (*R* >> 0.8). They confirm both micellar parameters as lipophilicity descriptors of the compounds that were tested.

### 3.1. QRARs

In QRARs, it is desirable to have the methodology relationships between solute retention and the biological activity of the compounds. In our investigations, we obtained the linear relationships between the micellar parameters (log *k*_m_ and log *K*_AM_) and the other parameters, i.e., log *K*_sc_ and log *P*_w, HSA_ (Figure 2) and *V*_d_, log *P*_w/pc_, *f*_u, brain_, and *f*_u, plasma_ (Figure 3) with very good quality (*R* >> 0.8). The straight lines in Figure 2A,B show a clear increase in dermal absorption and partitioning in the water–human serum albumin system of tested compounds with an increase in their lipophilicity. It should be noted that the lipophilic properties of the tested substances, based on the in silico log *P* parameters, are in the range of 5.386–1.345. The increase in lipophilicity in the parabolic function affects the volume of distribution (*V*_d_) and absorption by the plant cuticle (log *P*_w/pc_) as well as the unbonded fraction in the brain (*f*_u, brain_) and plasma (*f*_u, plasma_) (Figure 3). Although *V*_d_ and log *P*_w/pc_ increase with lipophilicity, the other parameters, i.e., *f*_u, plasma_ and *f*_u, brain_, decrease. The graphs suggest the existence of the optimal range of lipophilicity of the substance, for which the volume of distribution and the absorption through the epidermis are the highest, and the unbonded fractions in plasma and in the brain are the lowest. Figure 2 and Figure 3 also indicate that lipophilicity is the dominant factor that influences (1) the absorption of the test substances through the skin and epidermis, (2) the distribution of water–albumin, (3) the size of the unbound fraction in the plasma and the brain, and (4) the volume of the distribution.

### 3.2. QSARs

In the QSARs methodology, we used the experimentally-derived lipophilicities (micellar parameters log *k*_m_ and log *K*_AM_), and the numbers of hydrogen bond donors (*HBD*), acceptors (*HBA*), and rotatable bonds (*NRB*) as independent variables. These values were used to predict the dependent variables, i.e., log *K*_p_, log *K*_w/cell_, log *K*_HSA_, log *BB*, and *LD*_50_. Table 3 shows the quantitative structure–activity relationships (expressed as Equations (3)–(12)) that were established. The equations were cross-validated (LOO), and all of the calculated statistics are summarized in Table 3 and presented graphically in Figure 4, Figure 5, Figure 6, Figure 7 and Figure 8 as PLS-standardized coefficients (A), the response plots (B), and standardized residuals vs. leverages (C). The statistical parameters allowed us to positively evaluate the derived QSAR equations. There were no significant cross-correlations between the independent variables, and the values of the variance inflation factor (*VIF*) were significantly lower than 5. The diagrams presented in Figure 4A, Figure 5A, Figure 6A, Figure 7A and Figure 8A show the standard coefficients of Equations (3)–(12), and they explain the direction and the strength of the impact of a given descriptor on the calculated parameters. The correlations shown in Figure 4B, Figure 5B, Figure 6B, Figure 7B and Figure 8B illustrate the relationships between the actual response (values obtained from ACD/Percepta software) and those predicted by the established QSAR models (calculated response). The applicability domains (*AD*) of the developed regression models were also evaluated and are visualized as the Williams plots (Figure 4C, Figure 5C, Figure 6C, Figure 7C and Figure 8C). *AD* is a theoretical region in physicochemical space (the response and chemical structure space) for which a model should make predictions with a given reliability [45]. The warning leverage limits (*h** = 1.0) were calculated using the following equation:
molecules-27-03599-t003_Table 3Table 3The established quantitative structure–activity relationships: *n*—number of observations, *R*—coefficient of determination, *R*_adj._—adjusted coefficient of determination, *sd*—standard deviation, *F*-value, *p*—probability value, *VIF*—variance inflation factor, *PRESS*—predicted residual sum of squares, *MSE*—mean square error, cv—cross-validated.No.Equation(3)**log *K*_p_ = −5.329(0.156) + 0.597(0.041)log *k*_m_ − 0.669(0.075)*HBD* − 0.345(0.061)*HBA* + 0.197(0.044)*NRB****n* = 15; *R* = 0.9893; *R*_adj._ = 0.9850; *sd* = 0.110; *PRESS* = 0.276; *MSE* = 0.012; *F* = 116; *p* < 0.0001; *VIF* < 3.6; *PRESS*_cv_ = 0.276; *MSE*_cv_ = 0.012(4)**log *K*_p_ = −4.900(0.218) + 0.552(0.060)log *K*_AM_ − 0.714(0.118)*HBD* − 0.378(0.092)*HBA* + 0.213(0.067)*NRB****n* = 15; *R* = 0.9747; *R*_adj._ = 0.9644; *sd* = 0.169; *PRESS* = 0.610; *MSE* = 0.0286; *F* = 48; *p* < 0.0001; *VIF* < 3.5; *PRESS*_cv_ = 0.610; *MSE*_cv_ = 0.0286(5)**log *K*_w/cell_ = −2.773(0.295) + 0.062(0.078)log *k*_m_ − 1.846(0.143)*HBD* − 0.368(0.115)*HBA* + 0.076(0.083)*NRB****n* = 15; *R* = 0.9740; *R*_adj_ = 0.9634; *sd* = 0.209; *PRESS* =4.064; *MSE* =0.0435; *F* = 47; *p* < 0.0001; *VIF < 3.6*; *PRESS*_cv_ = 4.064; *MSE*_cv_ = 0.0435(6)**log *K*_w/cell_ = −2.762(0.262) + 0.079(0.073)log *K*_AM_ − 1.867(0.142)*HBD* − 0.356(0.111)*HBA* + 0.064(0.081)*NRB****n* = 15; *R* = 0.9753; *R*_adj_ = 0.9652; *sd* = 0.203; *PRESS* = 4.889; *MSE* =0.0414; *F* = 49; *p* < 0.0001; *VIF < 3.5*; *PRESS*_cv_ = 4.889; *MSE*_cv_ = 0.0400(7)**log *K*_HSA_ = 4.619(0.350) + 0.251(0.093)log *k_m_* + 0.106(0.170)*HBD* − 0.381(0.136)*HBA* + 0.150(0.100)*NRB****n* = 15; *R* = 0.8953; *R*_adj_ = 0.8498; *sd* = 0.248; *PRESS* = 1.078; *MSE* =0.0615; *F* = 11; *p* < 0.0001; *VIF < 3.6*; *PRESS*_cv_ = 1.078; *MSE*_cv_ = 0.0615(8)**log *K*_HSA_ = 4.843(0.351) + 0.204(0.097)log *K*_AM_ +0.109(0.190)*HBD* − 0.415(0.148)*HBA* + 0.175(0.108)*NRB****n* = 15; *R* = 0.8722; *R*_adj_ = 0.8155; *sd* = 0.272; *PRESS* = *1.326*; *MSE* =0.0741; *F* = 8; *p* < 0.0001; *VIF < 3.5*; *PRESS*_cv_ = 1.326; *MSE*_cv_ = 0.0741(9)**log *BB* = −0.375(0.078) + 0.254(0.021)log *k*_m_ − 0.454(0.038)*HBD* + 0.041(0.030)*HBA* − 0.006(0.023)*NRB****n* = 15; *R* = 0.9836; *R*_adj_ = 0.9770; *sd* = 0.055; *PRESS* = 0.090; *MSE* = 0.0031; *F* = 75; *p* < 0.0001; *VIF < 3.6*; *PRESS*_cv_ = 0.090; *MSE*_cv_ = 0.0031(10)**log *BB* = −0.205(0.073) + 0.242(0.020) log *K*_AM_ − 0.479(0.039) *HBD* + 0.032(0.031) *HBA* − 0.003(0.022) *NRB****n* = 15; *R* = 0.9831; *R*_adj_ = 0.9762; *sd* = 0.056; *PRESS* = 0.069; *MSE* = 0,0032; *F* = 72; *p* < 0.0001; *VIF < 3,5*; *PRESS*_cv_ = 0,069; *MSE*_cv_ = 0.0032(11)***LD*****_50_****= 2870(411) − 253(109) log *k*_m_ + 650(199) *HBD* − 194(160) HBA − 245(116) *NRB****n* = 15; *R* = 0.9330; *R*_adj_ = 0.9048; *sd* = 291; *PRESS* = 2078292; *MSE* = 84613; *F* = 17; *p* < 0.0001; *VIF < 3.6*; *PRESS*_cv_ = 2078292; *MSE*_cv_ = 76083(12)***LD*****_50_****= 2714(367) − 251(102) log *K*_AM_ +682(192) *HBD* − 192(155) *HBA* − 242(113) *NRB****n* = 15; *R* = 0.9358; *R*_adj_ = 0.9088; *sd* = 285; *PRESS* = 2017628; *MSE* = 81170; *F* = 18; *p* < 0.0001; *VIF < 3.5*; *PRESS*_cv_ = 2017628; *MSE*_cv_ = 74652
(13)$$h^{*}=\frac{3(k+1)}{n}$$
where *k* is the number of descriptors used in the MLR model, and *n* is the number of compounds in the data set. The Williams plot can be used for graphical detection of outliers (*h* > *h**).

The results proved that the models obtained are valid within the domain in which they were developed. The results obtained in our studies indicate the positive effect of solute lipophilicity on the skin (log *K*_p_) (Figure 4 and Figure 5) and cell permeation (log *K*_w/cell_) (Figure 6 and Figure 7) from water, binding affinity to human serum albumin (log *K*_HSA_) (Figure 8 and Figure 9), concentration in the brain (log *BB*) (Figure 10 and Figure 11), and toxicity in mice (the decrease in *LD*_50_) (Figure 12 and Figure 13). Lipophilicity is a dominant factor for log *K*_p_, log *K*_HSA_, log *BB*, and *LD*_50_. The rates of cell permeation are strongly retarded by solute hydrogen bond acidity and rather less so by hydrogen bond basicity (Figure 6 and Figure 7). The same effects of the compounds’ acidity and basicity on skin permeation were observed (Figure 4 and Figure 5). The number of hydrogen bond donors (*HBD)* also strongly reduces the substance permeation through the blood–brain barrier (Figure 10 and Figure 11) and increases the value of the lethal dose (Figure 12 and Figure 13). The values of *LD*_50_ decrease and the toxicity of the solutes increase with the number of hydrogen bond acceptors. Binding to human serum albumin is strongly related to (decreased) hydrogen bond basicity (*HBA*) and much less dependent (increased) on its acidity (*HBD*) (Figure 8 and Figure 9).

Solute flexibility, as described by the *NRB* values, strongly increases the rate of dermal absorption (Figure 4 and Figure 5) and binding to human serum albumin (Figure 8 and Figure 9). It also reduces the *LD*_50_ value, i.e., increases the toxicity of the substance (Figure 12 and Figure 13). *NRB* has a slightly positive effect on cell permeation (Figure 6 and Figure 7). Hydrogen bond basicity and solute flexibility practically do not affect the penetration of substances through the blood–brain barrier (Figure 10 and Figure 11).

When analyzing the results, substances no. 10–15 (phenoxyacetic acid derivatives) should be indicated as the most toxic for mice, i.e., having the lowest lethal dose after oral administration. These substances are more lipophilic among those tested (log *P* values are in the range of 2.6–5.386, with smaller *HBD* (*HBD* ≤ 1), and greater *HBA* (*HBA* ≥ 3) values, and they have the greatest number of rotatable bonds (*NRB* > 3). They also have a higher concentration in the brain; with the exception of compound no. 14, all of the log *BB* values were greater than 0.

Summarizing the results, substances no. 12 and no. 13 can be indicated as the most interesting among those that were tested. They are the most toxic, but they are also highly bound to plasma albumin, and their free fractions in plasma and the brain are the lowest. The magnitudes of the distribution are acceptable, as they were for all of the substances that were tested. On the basis of the results that were obtained, it can be concluded that they can be considered promising pesticides as well as subjects for further, more detailed research.

## 4. Materials and Methods

### 4.1. Chromatographic Measurements

Potential pesticides, i.e., carbamic (group I) and phenoxyacetic (group II) acids derivatives (Table 1), synthesized in our laboratory, were investigated using the micellar liquid chromatography technique. Previously, we reported [23] the chromatographic results that were obtained by the HPLC technique using a Purospher RP-8e column, four mobile phases composed of a buffer (pH = 7.4), four different sodium dodecyl sulfate (SDS) concentrations (i.e., 0.04, 0.06, 0.08, and 0.10 mol L^−1^), and the same 20% (*v*/*v*) addition of acetonitrile.

### 4.2. In Silico Parameters

The physicochemical, structural, pharmacokinetic, and toxicity parameters of the compounds that were tested were calculated from their molecular structures using ACD/Percepta software, version 1994-2012 (ACD/Labs, Advanced Chemistry Development, Inc., Toronto, ON, Canada) (Table 2).

### 4.3. Statistics

Linear regression (LR), multiple linear regression (MLR), partial last squares (PLS), and leave-one-out cross-validation (LOO) were conducted using the statistical software Minitab 16.2.4.0, version 1991-2004 (Minitab Inc., State College, PA, USA).

## 5. Conclusions

QRARs and QSAR methodologies were successful in modeling the pharmacokinetic properties and toxicities of 15 newly synthesized compounds considered as potential pesticides. The micellar liquid chromatography technique was used to determine the lipophilicity descriptors (log *k*_m_ and log *K*_AM_) of the compounds. In the QSAR method, log *k*_m_ and log *K*_AM_ parameters, *HBD*, *HBA*, and *NRB* were applied as independent values. All of the equations that were derived were evaluated statistically as being very good. The QSAR models that were developed had high predictive ability and high reliability in modeling the properties of the molecules that were tested. The investigations highlighted the significance and possibilities of combined chromatographic techniques and QR(S)ARs in modeling the important properties of potential pesticides and reducing unethical animal testing.

## Figures and Tables

**Figure 1 molecules-27-03599-f001:**
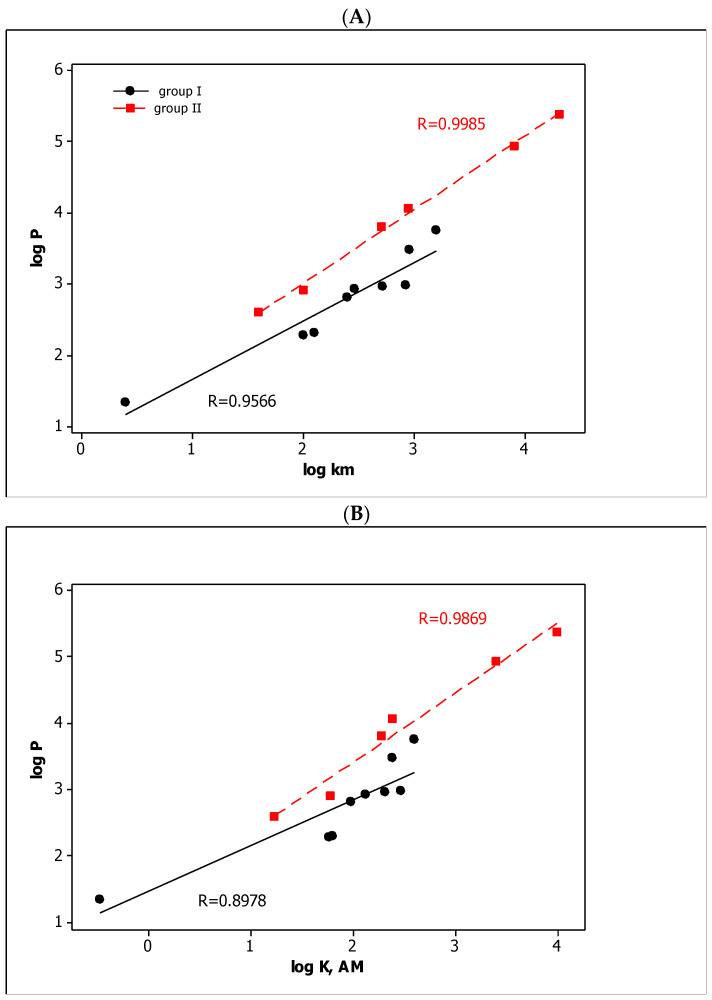
The log *P* vs. log *k*_m_ (**A**) and log *P* vs. log *K*_AM_ (**B**) relationships obtained for tested compounds.

**Figure 2 molecules-27-03599-f002:**
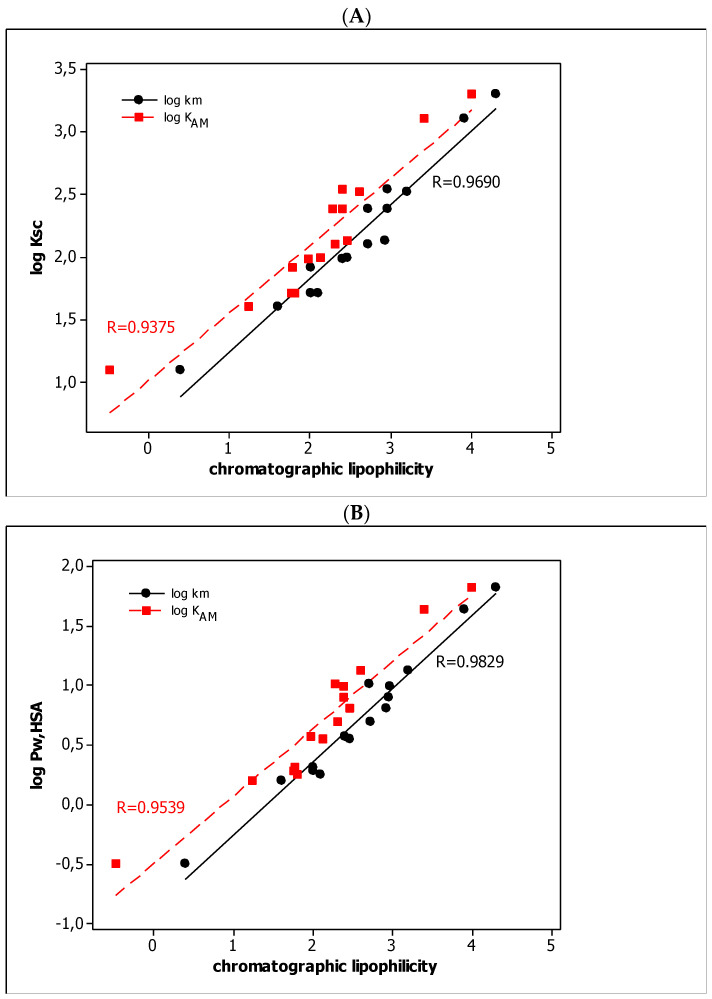
The linear relationships between log *K*_sc_ (**A**) and log P_w, HSA_ (**B**) and chromatographic parameters (log *k*_m_ and log *K*_AM_).

**Figure 3 molecules-27-03599-f003:**
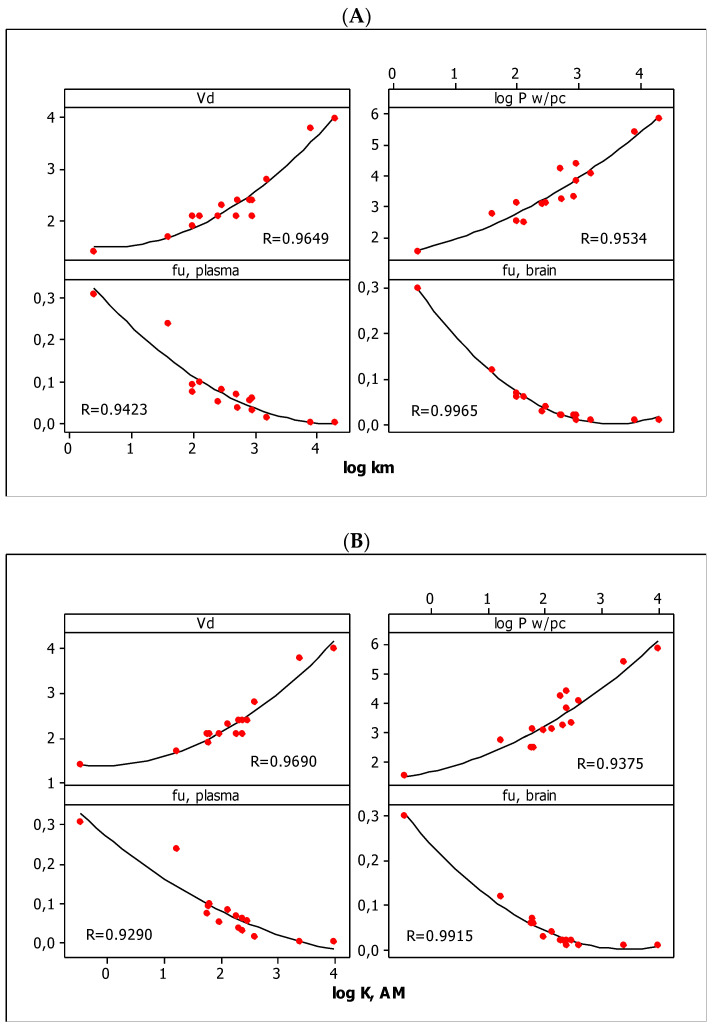
The parabolic relationships between pharmacokinetic parameters and log *k*_m_ (**A**) and log *K*_AM_ (**B**) values.

**Figure 4 molecules-27-03599-f004:**
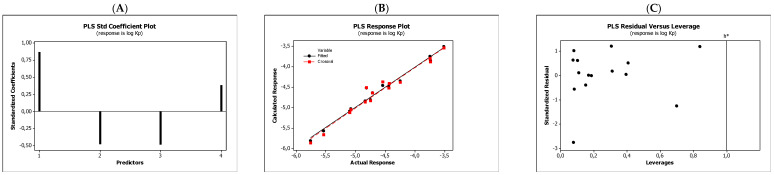
Standardized coefficients (**A**), the correlation between actual and predicted log *K*_p_ parameters (**B**), and the Williams plots (**C**) of Equation (3).

**Figure 5 molecules-27-03599-f005:**
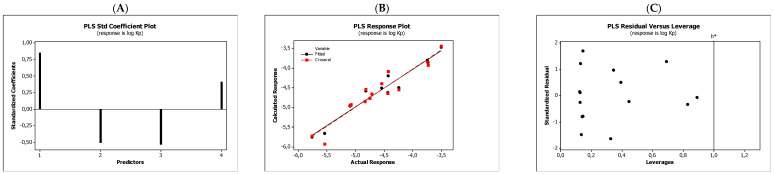
Standardized coefficients (**A**), the correlation between actual and predicted log *K*_p_ parameters (**B**), and the Williams plots (**C**) of Equation (4).

**Figure 6 molecules-27-03599-f006:**
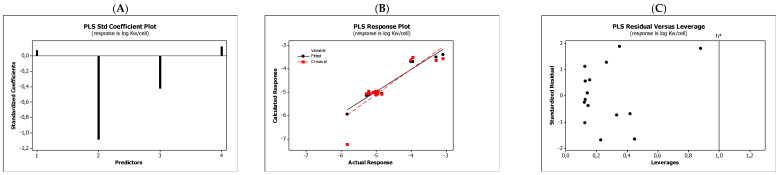
Standardized coefficients (**A**), the correlation between actual and predicted log *K*_w/cell_ parameters (**B**), and the Williams plots (**C**) of Equation (5).

**Figure 7 molecules-27-03599-f007:**
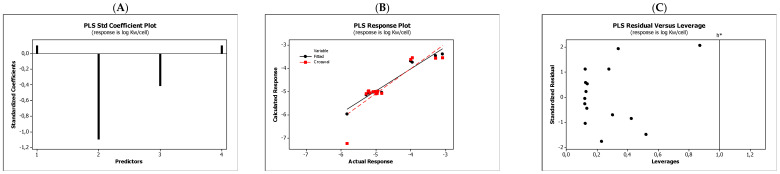
Standardized coefficients (**A**), the correlation between actual and predicted log *K*_w/cell_ parameters (**B**), and the Williams plots (**C**) of Equation (6).

**Figure 8 molecules-27-03599-f008:**
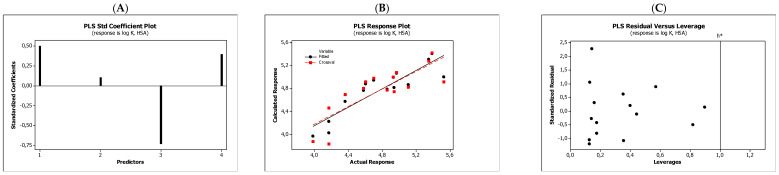
Standardized coefficients (**A**), the correlation between actual and predicted log *K*_a, HSA_ parameters (**B**), and the Williams plots (**C**) of Equation (7).

**Figure 9 molecules-27-03599-f009:**
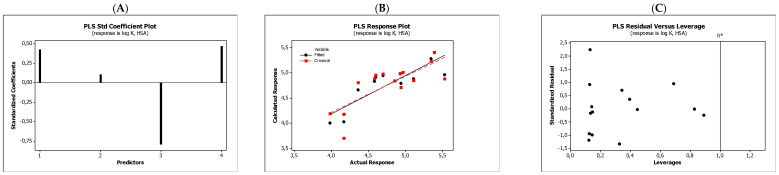
Standardized coefficients (**A**), the correlation between actual and predicted log *K*_a, HSA_ parameters (**B**), and the Williams plots (**C**) of Equation (8).

**Figure 10 molecules-27-03599-f010:**
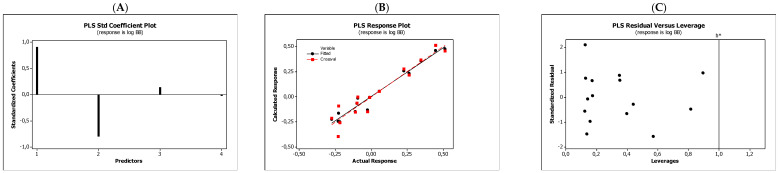
Standardized coefficients (**A**), the correlation between actual and predicted log *BB* parameters (**B**), and the Williams plots (**C**) of Equation (9).

**Figure 11 molecules-27-03599-f011:**
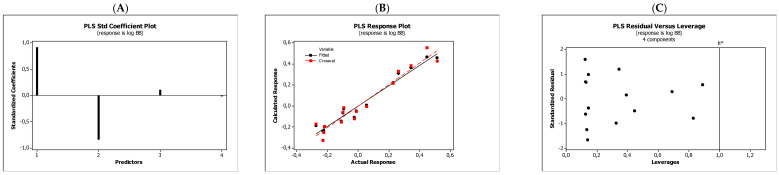
Standardized coefficients (**A**), the correlation between actual and predicted log *BB* parameters (**B**), and the Williams plots (**C**) of Equation (10).

**Figure 12 molecules-27-03599-f012:**
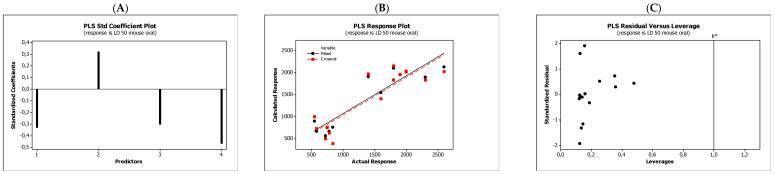
Standardized coefficients (**A**), the correlation between actual and predicted *LD*_50_ mouse oral parameters (**B**), and the Williams plots (**C**) of Equation (11).

**Figure 13 molecules-27-03599-f013:**
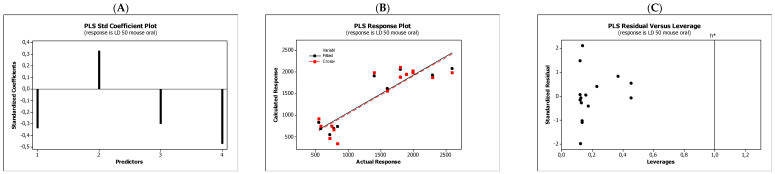
Standardized coefficients (**A**), the correlation between actual and predicted *LD*_50_ mouse oral parameters (**B**), and the Williams plots (**C**) of Equation (12).

**Table 1 molecules-27-03599-t001:** Structures and chromatographic parameters [23] of tested compounds.

No	Group	Structure	log *k*_m_	log *K*_AM_
1		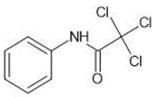	2.00	1.76
2		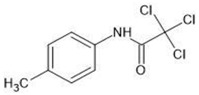	2.46	2.13
3		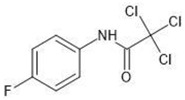	2.10	1.80
4		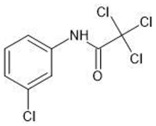	2.72	2.31
5	I	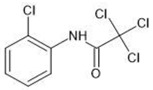	2.40	1.98
6		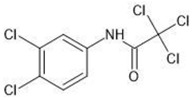	3.20	2.60
7		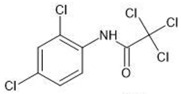	2.96	2.39
8		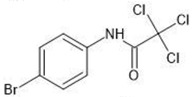	2.92	2.46
9		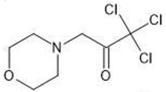	0.40	−0.48
10		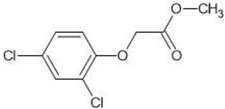	2.00	1.76
11		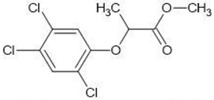	2.95	2.39
12		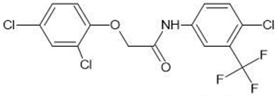	3.90	3.40
13	II	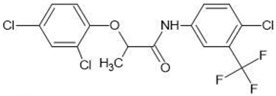	4.30	4.00
14		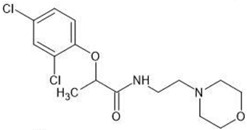	1.60	1.23
15		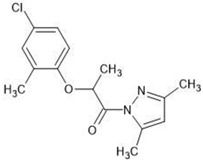	2.71	2.28

**Table 2 molecules-27-03599-t002:** Physicochemical, pharmacokinetic, and toxicity parameters of tested compounds.

Nr	log *P*	*HBD*	*HBA*	*NRB*	*MW* [g/mol]	*TPSA* [A^2^]	*V*_d_ [L/kg]	*f* _u, plasma_	*f* _u, brain_	log *BB*	log *K*_p_	log *K*_sc_	log *K*_w/cell_	log *P*_w/HSA_	log *K*_HSA_	log *P*_w/pc_	*LD*_50_ [mg/kg]
1	2.285	1	2	2	238.50	29.10	2.1	0.0750	0.06	−0.218	−5.095	1.713	−4.988	0.280	4.58	2.513	2600
2	2.937	1	2	2	252.52	29.10	2.3	0.0830	0.04	−0.026	−4.747	1.996	−4.833	0.554	4.60	3.106	2000
3	2.309	1	2	2	256.49	29.10	2.1	0.1000	0.06	−0.275	−5.075	1.717	−5.164	0.253	4.86	2.504	1800
4	2.971	1	2	2	272.94	29.10	2.4	0.0370	0.02	−0.098	−4.710	2.100	−5.071	0.694	4.70	3.254	1900
5	2.810	1	2	2	272.94	29.10	2.1	0.0510	0.03	−0.110	−4.836	1.989	−4.951	0.576	5.11	3.077	2000
6	3.755	1	2	2	307.39	29.10	2.8	0.0140	0.01	0.056	−4.244	2.529	−4.944	1.132	4.97	4.092	1800
7	3.488	1	2	2	307.39	29.10	2.4	0.0320	0.01	−0.011	−4.434	2.390	−5.020	0.993	5.53	3.847	2300
8	2.980	1	2	2	317.39	29.10	2.4	0.0540	0.02	−0.094	−4.818	2.134	−5.197	0.804	4.93	3.331	1400
9	1.345	0	3	3	232.49	29.54	1.5	0.3100	0.30	−0.225	−5.538	1.097	−3.947	−0.492	4.17	1.522	1600
10	2.917	0	3	4	235.06	35.53	1.9	0.0920	0.07	0.263	−4.431	1.923	−3.279	0.313	4.36	3.132	550
11	4.067	0	3	4	283.53	35.53	2.1	0.0610	0.02	0.513	−3.731	2.541	−3.081	0.904	4.94	4.400	580
12	4.936	1	3	5	398.59	38.33	3.8	0.0030	0.01	0.227	−3.741	3.107	−5.275	1.642	5.35	5.420	780
13	5.386	1	3	5	412.62	38.33	4.0	0.0029	0.01	0.345	−3.504	3.308	−5.213	1.823	5.39	5.858	720
14	2.600	1	5	5	347.24	50.80	1.6	0.2400	0.12	−0.229	−5.761	1.608	−5.822	0.200	3.98	2.765	840
15	3.807	0	4	3	292.76	44.12	2.1	0.0690	0.02	0.448	−4.540	2.386	−3.999	1.017	4.17	4.240	750

## Data Availability

Not applicable.

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
