# Peer review of "Quantitative Retention (Structure)–Activity Relationships in Predicting the Pharmaceutical and Toxic Properties of Potential Pesticides"

_molecules, 2022, doi:10.3390/molecules27113599_

Round 1
Reviewer 1 Report
Comments on the manuscript: “Quantitative Retention (Structure)-Activity Relationships in Predicting the Pharmaceutical and Toxic Properties of Potential.
Pesticides”
[Manuscript ID: molecules-1720217]
In this manuscript, the authors have reported a new quantitative retention (structure)-activity relationships method for predicting the pharmaceutical and toxic properties of potential. The novelty and methodology of the article are adequate well-written. Therefore, it suitable for publication in "Molecules " journal. Some minor comments are as follows:
- Write keywords in alphabetic order.
- Define log Ksc in section 3.1. QRARs
Author Response
Point 1: Write keywords in alphabetic order.
Response 1: Done.
Point 2: Define log Ksc in section 3.1. QRARs
Response 2: The definition of log Ksc parameter (i.e. skin-water partition coefficient) has been added (line 109).
Reviewer 2 Report
In this study, QRARs and the two-dimensional QSAR methodology were successfully used in evaluating the pharmacokinetic properties and toxicities of 15 newly-synthesized compounds which considered as potential pesticides. The developed QSAR models had high predictive ability and high reliability in modeling the properties of the molecules that were tested. The investigations highlighted the significance and possibilities of combined chromatographic techniques and QR(S)ARs in modeling the important properties of potential pesticides and reducing unethical animal testing. However, the manuscript should be addressed following issues and revised in detail before reconsidering acceptance for publication.
- Table 1 and Table 2 mentioned in this manuscript cannot be found in the manuscript, and the relevant tables mentioned should be listed in it.
- The data plot analysis is messy and not thorough, and the authors should analyze them in an orderly and detailed manner.
- The structural framework of this manuscript is confusing, and "materials and methods" should be placed in the first half of the manuscript, besides the experimental process should be explained.
Author Response
Point 1: Table 1 and Table 2 mentioned in this manuscript cannot be found in the manuscript, and the relevant tables mentioned should be listed in it.
Response 1: All tables mentioned in the manuscript (Tables 1-3) were inserted by us into the system (Submissions Menu). We cannot explain why they were not sent to Reviewers. Our guess is that they have not been combined into the PDF file with text and drawings. We sincerely apologize for this error and attach tables for inspection: Table 1 contains the structures and chromatographic parameters of tested compounds, Table 2 - their physicochemical, pharmacokinetic, and toxicity parameters, and Table 3 - the established QSA relationships (i.e. equations 3-12);
Point 2: The data plot analysis is messy and not thorough, and the authors should analyze them in an orderly and detailed manner.
Response 2: Reviewer’s remark that data analysis is “messy and not thorough” seems unfair and perhaps exaggerated. In the QSARs procedure, it is important to indicate those properties of substances that have the most important influence on the activity of the substance, in our case, on the pharmacokinetics and toxicity. Therefore, the obtained results were analyzed by assessing the influence of lipophilicity (chromatographic parameters log km and log KAM), the number of hydrogen bond donors (HBD), and acceptors (HBA), and rotatable bonds (NRB) on selected parameters (log Kp, log BB, etc.). Due to the multitude of pharmacokinetic parameters, the analysis may appear somewhat confusing. In the case of the other Reviewers, both the order and the accuracy of the discussion did not raise any doubts. Nevertheless, respecting the opinion of the Reviewer, we have made an amendment to section 3.2, which explains the plan for analyzing the results.
Point 3: The structural framework of this manuscript is confusing, and "materials and methods" should be placed in the first half of the manuscript, besides the experimental process should be explained.
Response 3: The structural framework of this manuscript complies with the requirements of the MDPI (see instruction for authors).
Reviewer 3 Report
To whom it may concern: the article "Quantitative Retention (Structure)-Activity Relationships in Predicting the Pharmaceutical and Toxic Properties of Potential Pesticides" by the authors M. Janicka and A. Śliwińska reports about the prediction of the properties of novel potential pesticides by application of QSAR and micellar chromatography techniques. The article is clear and well written. There are however some comments which need to be addressed prior to article acceptance:
Major remarks:
Any work including QSAR methodology should necessarily depict the structures of the investigated compounds. Where are these? The Table 1 (referred to in the line 87) is completely missing from the article, as well as Table 2. Please include these in the revised version of the article.
The equations 3-12, first referred to in the line 183, are also missing from the article. Include these in the revised version of the article.
-line123: did you perhaps mean "g/mol" instead of "mg/mole"?
Minor remarks:
-line 17: "a functions" remove "a"
-line 23: instead of "significance and possibilities" it is more appropriate to write "applicability"
-line 32: instead of "exposed to constant contact with" it is better to write "constantly exposed to"
-line 50: instead of "offer" it should say "are offered by"
-line 55: instead of "hydro-" it should say "aqueous-"
-line85: instead of "included" it should say "present"
-line91: remove "a" from "a functions"
-line105: change "n" in "n-octanol" to italic
-line116: change "possibilities" to "application"
-line123: change "A2" to "Å2"
-line135: Are small values of fraction bound to serum albumin really always preferable, and is that with regard to unintentionally exposed humans or unwanted pests?
-line151: see the comment for line 105
-line 226: instead of "smaller acidity" and "greater basicity" it is more appropriate to say "smaller number of hydrogen bond donors" and "larger number of hydrogen bond acceptors"
-line255: remove "two-dimensional"
-line256: "newly synthesized" should have no hyphen
With best regards
Author Response
Major remarks:
Point 1: Any work including QSAR methodology should necessarily depict the structures of the investigated compounds. Where are these? The Table 1 (referred to in the line 87) is completely missing from the article, as well as Table 2. Please include these in the revised version of the article.
Response 1: All tables mentioned in the manuscript (Tables 1-3) were inserted by us into the system (Submissions Menu). We cannot explain why they were not sent to Reviewers. Our guess is that they have not been combined into the PDF file with text and drawings. We sincerely apologize for this error and attach tables for inspection: Table 1 contains the structures and chromatographic parameters of tested compounds, Table 2 - their physicochemical, pharmacokinetic, and toxicity parameters, and Table 3 - the established QSA relationships (i.e. equations 3-12).
Point 2: The equations 3-12, first referred to in the line 183, are also missing from the article. Include these in the revised version of the article.
Response 2: The equations 3-12 are presented in Table 3 (see Response 1)
Point 3: line 123: did you perhaps mean "g/mol" instead of "mg/mole"?
Response 3: of course it should be "g/mol" and has been corrected. (You were the only one to notice it:)).
Minor remarks:
-line 17: "a functions" remove "a"
Response: done
-line 23: instead of "significance and possibilities" it is more appropriate to write "applicability"
Response: done
-line 32: instead of "exposed to constant contact with" it is better to write "constantly exposed to"
Response: done
-line 50: instead of "offer" it should say "are offered by"
Response: done
-line 55: instead of "hydro-" it should say "aqueous-"
Response: done
-line 85: instead of "included" it should say "present"
Response: done
-line 91: remove "a" from "a functions"
Response: done
-line 105: change "n" in "n-octanol" to italic
Response: done
-line116: change "possibilities" to "application"
Response: done
-line 123: change "A2" to "Å2"
Response: done
-line 135: Are small values of fraction bound to serum albumin really always preferable, and is that with regard to unintentionally exposed humans or unwanted pests?
Response: the sentence: “The binding to serum albumins affects the concentration of the unbonded forms of the substance in serum, and small values are preferable in order to prevent possible side effects” included in the manuscript means exactly what the reviewer said in the opinion. To be clear, an amendment was introduced: “The binding to serum albumins affects the concentration of the unbonded forms of the substance in serum, and small values of free fractions are preferable in order to prevent possible side effects.”;
-line 151: see the comment for line 105
Response: done
-line 226: instead of "smaller acidity" and "greater basicity" it is more appropriate to say "smaller number of hydrogen bond donors" and "larger number of hydrogen bond acceptors"
Response: done
-line 255: remove "two-dimensional"
Response: done
-line 256: "newly synthesized" should have no hyphen
Response: done

Round 2
Reviewer 2 Report
Structures in Table 1 should be redrawed using chemioffice software.
Author Response
- Structures in Table 1 should be redrawed using chemioffice software. Responce: done
Reviewer 3 Report
To whom it may concern: the article "Quantitative Retention (Structure)-Activity Relationships in Predicting the Pharmaceutical and Toxic Properties of Potential Pesticides" by the authors M. Janicka and A. Śliwińska can be accepted for publication.
Prior to acceptance, please change the abbreviation for median lethal dose throughout the text from "LD 50" to "LD50".
With best regards
Author Response
Prior to acceptance, please change the abbreviation for median lethal dose throughout the text from "LD 50" to "LD50".
Response: done
Sincerely Yours